# Evolution of the gene regulatory network of body axis by enhancer hijacking in amphioxus

Chenggang Shi, Shuang Chen, Huimin Liu, Rongrong Pan, Shiqi Li, Yanhui Wang, Xiaotong Wu, Jingjing Li, Xuewen Li, Chaofan Xing, Xian Liu, Yiquan Wang*, Qingming Qu*, Guang Li*

State Key Laboratory of Cellular Stress Biology, School of Life Sciences, Xiamen University, Xiamen, China

**Abstract** A central goal of evolutionary developmental biology is to decipher the evolutionary pattern of gene regulatory networks (GRNs) that control embryonic development, and the mechanism underlying GRNs evolution. The Nodal signaling that governs the body axes of deuterostomes exhibits a conserved GRN orchestrated principally by Nodal, Gdf1/3, and Lefty. Here we show that this GRN has been rewired in cephalochordate amphioxus. We found that while the amphioxus *Gdf1/3* ortholog exhibited nearly no embryonic expression, its duplicate *Gdf1/3-like*, linked to *Lefty*, was zygotically expressed in a similar pattern as *Lefty*. Consistent with this, while *Gdf1/3-like* mutants showed defects in axial development, *Gdf1/3* mutants did not. Further transgenic analyses showed that the intergenic region between *Gdf1/3-like* and *Lefty* could drive reporter gene expression as that of the two genes. These results indicated that *Gdf1/3-like* has taken over the axial development role of *Gdf1/3* in amphioxus, possibly through hijacking *Lefty* enhancers. We finally demonstrated that, to compensate for the loss of maternal *Gdf1/3* expression, Nodal has become an indispensable maternal factor in amphioxus and its maternal mutants caused axial defects as *Gdf1/3-like* mutants. We therefore demonstrated a case that the evolution of GRNs could be triggered by enhancer hijacking events. This pivotal event has allowed the emergence of a new GRN in extant amphioxus, presumably through a stepwise process. In addition, the co-expression of *Gdf1/3-like* and *Lefty* achieved by a shared regulatory region may have provided robustness during body axis formation, which provides a selection-based hypothesis for the phenomena called developmental system drift.

**\*For correspondence:**
wangyq@xmu.edu.cn (YW);
quqingming@xmu.edu.cn (QQ);
guangli@xmu.edu.cn (GL)

**Competing interest:** The authors declare that no competing interests exist.

## Editor's evaluation

This valuable work advances our understanding of how deuterostome body plan is established during development as well as how the gene regulatory network governing early embryogenesis has been rewired during animal evolution. The evidence supporting the conclusions is compelling, with a suite of molecular-level experiments. The work will be of broad interest to developmental biologists.

## Introduction

The developmental regulatory mechanisms of body axis formation have been investigated in numerous metazoans, which have greatly advanced evolutionary developmental biology (evo-devo) as a research field (*Hall, 1999*). Importantly, these accumulated data also make it possible to analyze how the gene regulatory networks (GRNs) controlling body axes has been evolving in different clades, a key theme in evo-devo (*Peter and Davidson, 2011*; *Hall, 1999*). Nevertheless, detailed functional

genetic evidence showing how GRNs could be rewired during evolution is usually lacking, which hinders our understanding of the evolvability of organisms.

Nodal signaling plays a conserved role in patterning the dorsal–ventral (D–V) and left–right (L–R) axes in deuterostomes, exhibiting a conserved GRN orchestrated principally by Nodal, Gdf1/3, and Lefty (*Jia and Meng, 2021*). Different from most deuterostomes which has one *Gdf1/3* gene, mammals and frogs have two such genes (namely *Gdf1* and *Gdf3*), derived from lineage-specific duplications (*Opazo and Zavala, 2018*; *Opazo et al., 2019*). For simplicity, we collectively call them *Gdf1/3* genes in this study. The expression patterns of genes coding Nodal and Gdf1/3 are highly conserved in echinoderms and vertebrates, with *Nodal* being expressed zygotically (unilaterally at neurula or larva stage) (*Supplementary file 1a*), and *Gdf1/3* both maternally and zygotically (but bilaterally at neurula stage) (*Supplementary file 1b*). Zygotic Nodal functions synergistically with preexisting maternal Gdf1/3 by forming heterodimers to activate the signaling pathway (*Montague and Schier, 2017*; *Opazo et al., 2019*). Robust Nodal signaling is safeguarded by a positive-feedback loop from further activation of *Nodal* itself and a negative feedback loop through inducing the expression of *Lefty* encoding an inhibitor of the signaling (*Müller et al., 2012*).

Previous studies have identified one *Nodal*, one *Lefty* and two *Gdf1/3* (tentatively named *GDF1/3-like1/Vg1* and *GDF1/3-like2*) genes in basally divergent chordate amphioxus (*Yu et al., 2002*; *Satou et al., 2008*; *Onai et al., 2010*). The expression pattern of *GDF1/3-like2* has not been investigated before this study. Like the orthologs in echinoderms (*Duboc et al., 2008*) and vertebrates (*Meno et al., 1997*; *Bisgrove et al., 1999*; *Ishimaru et al., 2000*; *Tanegashima et al., 2000*), *Lefty* gene in amphioxus is expressed zygotically and unilaterally (*Onai et al., 2010*; *Soukup et al., 2015*; *Morov et al., 2016*; *Zhang et al., 2019*). Unusually, it was found that *Nodal* and *GDF1/3-like1/Vg1* are both maternally supplied and zygotically expressed unilaterally at the neurula stage (*Supplementary file 1a and b*). This implies that the regulatory network governing the Nodal signaling pathway has undergone alterations within this particular clade, while its function in D–V and L–R axes patterning has been preserved (*Onai et al., 2010*; *Soukup et al., 2015*; *Li et al., 2017*; *Zhang et al., 2019*). We thus speculate that the Nodal signaling of amphioxus would be an excellent case to trace the evolutionary history of a GRN and to clarify the mechanism underlying it. To evaluate this, we analyzed the GRN of amphioxus Nodal signaling using mutant and transgenic lines and dissected its evolutionary history by integrating available functional genetic data from other deuterostomes.

## Results

### Evolution of the two *Gdf1/3* genes in amphioxus

*Gdf1/3* genes have only been detected in deuterostomes (*Stenzel et al., 1994*; *Lapraz et al., 2006*; *Satou et al., 2008*; *Simakov et al., 2015*; *Opazo and Zavala, 2018*). The gene is linked to *Bmp2/4* in many deuterostome species (*Figure 1A*; *Range et al., 2007*; *Satou et al., 2008*), though this linkage is lost in higher vertebrates like mice, chickens, and frogs (*Opazo and Zavala, 2018*). Accordingly, sequence similarity and phylogenetic evidence supported the view that *Gdf1/3* originated from *Bmp2/4* by a tandem duplication event that occurred in the common ancestor of deuterostomes (*Figure 1A*, *Figure 1—figure supplement 1*; *Lapraz et al., 2006*; *Range et al., 2007*; *Satou et al., 2008*). However, Floridae amphioxus (*Branchiostoma floridae*) was reported to contain two *Gdf1/3* genes (*Satou et al., 2008*), one of which (previously named *GDF1/3-like2* by *Satou et al., 2008*) is linked to *Bmp2/4* as was found in other deuterostomes, representing the ancestral amphioxus *Gdf1/3* gene (hereafter renamed *Gdf1/3*, *Figure 1A*). Intriguingly, the other one (previously known as *GDF1/3-like1* [*Satou et al., 2008*] or *Vg1* [*Onai et al., 2010*]), here renamed as *Gdf1/3-like* (*Figure 1A*), is also linked to a transforming growth factor-β (TGF-β) family gene *Lefty* (*Satou et al., 2008*). This peculiar gene arrangement, together with that the searches of *Lefty* ortholog were positive only in deuterostomes until 2008, led Satou et al. proposing that the *Gdf1/3-like–Lefty* gene pair derived from either a duplication of the *Gdf1/3–Bmp2/4* gene pair or two duplications of *Gdf1/3* (*Gdf1/3* duplicated first to generate *Gdf1/3-like*, which then translocated and duplicated tandemly to generate *Lefty*) (*Satou et al., 2008*). However, this view was not supported by molecular phylogenetic analysis (*Satou et al., 2008*) and was rejected by the recent finding of *Lefty* genes in Lophotrochozoa (*Heger et al., 2020*). Further survey on recently updated genomes of bilaterians, especially those reported to have *Lefty* genes, revealed that the *Gdf1/3-like* gene and its linkage to *Lefty* exist only in amphioxus species, but

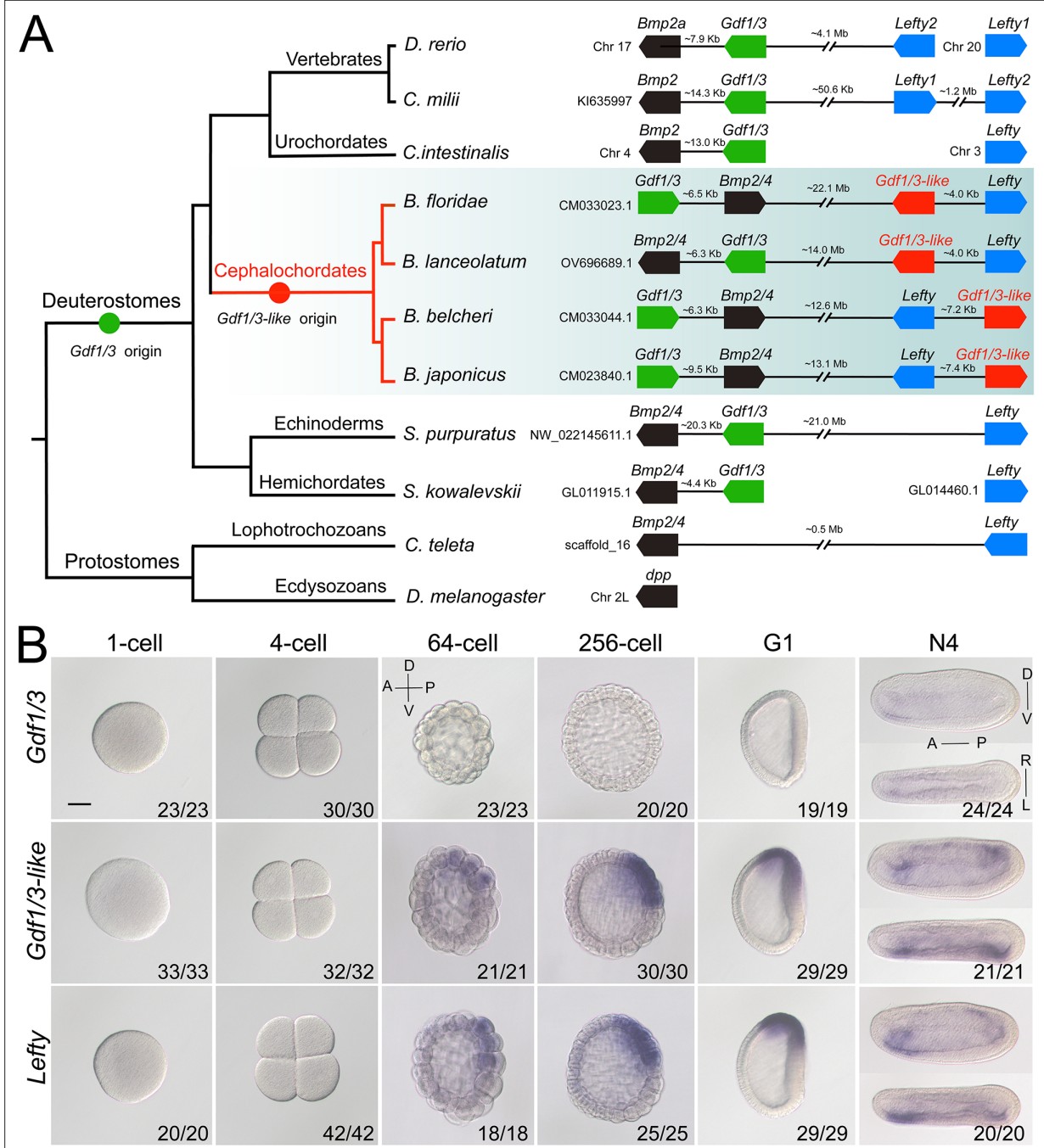

**Figure 1.** Synteny and expression pattern of amphioxus *Gdf1/3* and *Gdf1/3-like* gene. (**A**) Arrangement of *Gdf1/3*, *Bmp2/4*, *Gdf1/3-like*, and *Lefty* genes in representative bilaterian genomes. Black lines represent the genes (boxes) at both ends are tightly linked and fractured lines represent the genes are close together on a same chromosome or scafford indicated. zebrafish (*Danio rerio*), elephant shark (*Callorhinchus milii*), vase tunicate (*Ciona intestinalis*), Florida amphioxus (*B. floridae*), European amphioxus (*B. lanceolatum*), Asia amphioxus (*B. belcheri* and *B. japonicum*), sea urchin (*Strongylocentrotus purpuratus*), acorn worm (*Saccoglossus kowalevskii*), polychaete worm (*Capitella teleta*), and fruit fly (*Drosophila melanogaster*). (**B**) The spatiotemporal expression pattern of *Gdf1/3*, *Gdf1/3-like*, and *Lefty* at different stages of *B. floridae* embryos. Embryos at 64-cell to G1 stage were viewed from the left side with anterior to the left (the anterior-posterior [A–P] and D–V axes are labeled), and those at N4 stage were viewed from either the left site (upper panels) or dorsal side (lower panels) with anterior to the left (the A–P, D–V, and L–R axes are all labeled). Numbers in the panels indicate the number of times the expression pattern shown was identified, out of the total number of embryos identified. Scale bar, 50 μm.

The online version of this article includes the following source data and figure supplement(s) for figure 1:

**Figure supplement 1.** Phylogenetic tree of *Gdf1/3*-related genes.

*Figure 1 continued on next page*

*Figure 1 continued*

**Figure supplement 2.** Analysis of *Gdf1/3*, *Gdf1/3-like*, and *Lefty* expression patterns at different stages of *B. floridae* embryos or larvae with *in situ* hybridization.

**Figure supplement 3.** Analysis of *Gdf1/3*, *Gdf1/3-like*, and *Lefty* expression patterns at different stages of *B. floridae* embryos or larvae with quantitative real-time polymerase chain reaction.

**Figure supplement 3—source data 1.** The threshold cycle of each gene analyzed using quantitative real-time polymerase chain reaction.

**Figure supplement 4.** Analysis of *Nodal*, *Gdf1/3*, *Gdf1/3-like*, and *Lefty* expression patterns at different stages of *B. lanceolatum* embryos or larvae with RNA-seq.

**Figure supplement 5.** Analysis of *Gdf1/3-like* and *Lefty* expression patterns at different stages of *B. floridae* embryos with double fluorescence *in situ* hybridization.

not in any other bilaterians examined (*Figure 1A*). These findings suggest that the *Gdf1/3-like* gene most likely arose in Cephalochordata, or at least in the genus of *Brachiostoma*, through a tandem duplication of *Gdf1/3*, followed by a translocation of it to the *Lefty* locus (*Figure 1A*). In line with this proposal, lineage-specific duplication of the *Gdf1/3* gene and translocation of the duplicate to other genomic regions have also been found in at least two lineages of vertebrates (*Opazo and Zavala, 2018*; *Opazo et al., 2019*).

## Amphioxus *Gdf1/3* lost the ancestral role in body axes formation

We next asked if the *Gdf1/3* in amphioxus is functionally similar to its ortholog in vertebrates and sea urchins. We first analyzed its embryonic expression pattern using different methods (*in situ* hybridization, quantitative real-time polymerase chain reaction, and published RNA-seq data; *Marlétaz et al., 2018*). Unexpectedly, no *Gdf1/3* transcripts were detected in amphioxus embryos before the neurula stage, and very weak expression of the gene was found in few cells of the anterior ventral pharyngeal region during the late neurula and larva stages (*Figure 1B*, *Figure 1—figure supplements 2–4*). We further generated amphioxus mutants of the *Gdf1/3* gene. Consistent with its restricted and weak expression pattern, the homozygous *Gdf1/3* mutants (*Gdf1/3$^{-/-}$*) displayed normal D–V and L–R axis patterning as the wild-type (WT) by the 3-gill slit stage (*Figure 2—figure supplement 1*). These data suggested that *Gdf1/3* has disassociated from the GRN of body axis formation in living amphioxus. It is, however, notable that overexpression of *Gdf1/3* by injecting its mRNA could result in the expansion of anterior and dorsal identity (expressing *FoxQ2*, *Wnt3*, *Brachyury*, and *Chordin*) at the expense of ventral structures (expressing *Evx*) (*Figure 2—figure supplement 2*), as overactivation of the Nodal signaling (*Onai et al., 2010*; *Zhang et al., 2019*). Moreover, treatment with SB505124, a selective inhibitor of Alk4/5/7, could ventralize the embryos injected (*Figure 2—figure supplement 2*). These results demonstrated that the Gdf1/3 still works as a ligand of Nodal signaling, although it is dispensable to the body axis formation of extant amphioxus.

## *Gdf1/3-like* is indispensable for body axes formation in amphioxus

As *Gdf1/3* is no longer involved in the body axis formation in amphioxus, while injection of *Gdf1/3* and *Gdf1/3-like* mRNA yielded similar phenotypes (*Figure 2—figure supplement 2*), we hypothesized that *Gdf1/3-like* is involved in the body axes formation. To test this, we first reanalyzed the expression pattern of *Gdf1/3-like* in amphioxus using different methods (*in situ* hybridization, quantitative real-time polymerase chain reaction, and published RNA-seq data; *Marlétaz et al., 2018*). Inconsistent with previous study (*Onai et al., 2010*), no maternal expression of *Gdf1/3-like* was detected in our analyses (*Figure 1B*, *Figure 1—figure supplements 2–4*). Zygotic expression of the gene was first detected in the dorsal blastomeres of the vegetal pole at the 32- to 64-cell stage, and then restricted in the dorsal blastopore lip (Spemann organizer) at the gastrula stage and the left side of the embryo from the early neurula stage (*Figure 1B*, *Figure 1—figure supplement 2*). Notably, the expression pattern of *Gdf1/3-like* is similar (although not identical) to that of *Lefty* (*Figure 1B*, *Figure 1—figure supplements 2–5*).

We then created *Gdf1/3-like* mutants to see if it is required for amphioxus body axis development. *Gdf1/3-like$^{-/-}$* embryos showed severe defects in axis formation. At the G5 stage, the *Gdf1/3-like$^{-/-}$* embryos failed to flatten dorsally like the WT embryos (*Figure 2—figure supplement 3A*), and at subsequent stages they lacked most of the anterior and dorsal structures (*Figure 2A*, *Figure 2—figure*

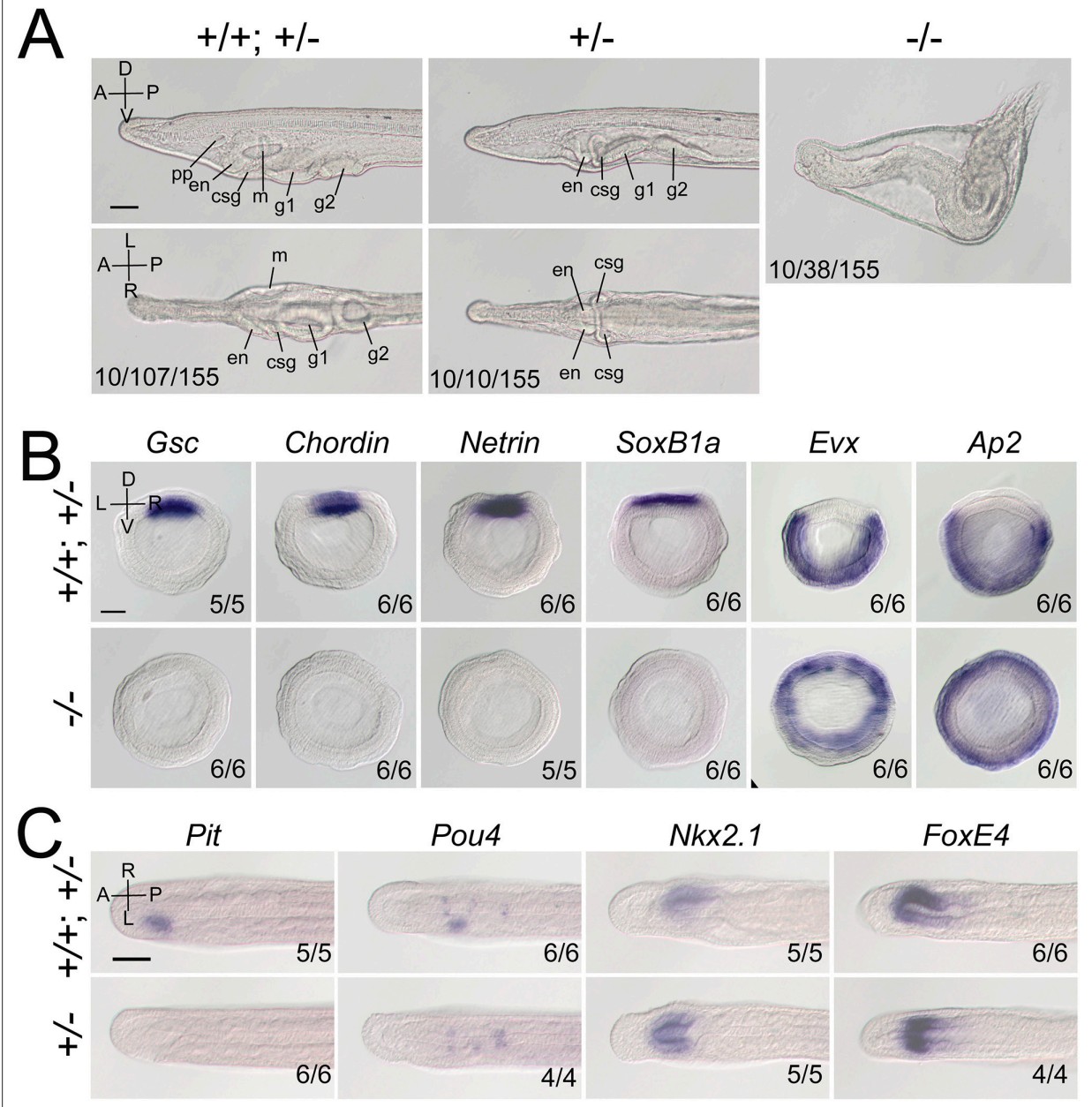

**Figure 2.** *Gdf1/3-like* loss-of-function affects amphioxus axes development. (**A**) Phenotypic analyses of *Gdf1/3-like* mutants. Larvae at L2 stage were observed from the left side (upper panels) and ventral side (under panels) with anterior to the left (the A–P, D–V, and L–R axes are labeled). pp, preoral pit; en, endostyle; csg, club-shaped gland; m, mouth; g1, first gill slit; g2, second gill slit. The phenotypes of embryos from a same pool at early stages are provided in *Figure 2—figure supplement 3*. The three numbers (from left to right) at the bottom right of each panel indicate the number of larvae used for genotyping, the number of larvae with the phenotype, and the total number of larvae examined, respectively. (**B, C**) The expression of marker genes in *Gdf1/3-like* mutants. Embryos in (**B**) were at G5 stage and viewed from the blastopore with dorsal to the top (the D–V and L–R axes are labeled), while embryos in (**C**) were at T0–T1 stage and viewed from the dorsal side with anterior to the left (the A–P and L–R axes are labeled). Numbers at the bottom right indicate the number of times the genotype shown was identified, out of the total number of examined embryos with the expression pattern. The nomenclature +/+; +/– refer to a pool of animals with different genotypes (+/+ and +/–) in the figures of present study. Scale bars, 50 μm.

The online version of this article includes the following figure supplement(s) for figure 2:

**Figure supplement 1.** Phenotypic analysis of amphioxus *Gdf1/3* mutants.

**Figure supplement 2.** The expression of marker genes in embryos injected with *Gdf1/3* or *Gdf1/3-like* mRNA and treated with dimethyl sulfoxide or SB505124.

**Figure supplement 3.** Phenotypic analysis of *Gdf1/3-like* mutants and the expression of marker genes in *Gdf1/3-like* mutants.

**Figure supplement 4.** The expression pattern of *Nodal* and *Gsc* in *Gdf1/3-like* mutants at G1 stage.

supplement 3A). In addition, by the L2 stage, there was a minority of *Gdf1/3-like*$^{+/-}$ larvae showing no left but two-right phenotype (*Figure 2A*). We further analyzed the expression patterns of the marker genes for various structures in the mutants. At the gastrulae (G5) stage of *Gdf1/3-like*$^{-/-}$, the expression of *Gsc* in dorsal mesoderm, *Chordin* and *Netrin* in dorsal mesoderm and neural ectoderm and *SoxB1a* in dorsal pan-neural ectoderm disappeared, while that of *Evx* in ventral domain and *Ap2* in epidermal ectoderm expanded dorsally (*Figure 2B*). At the T0 stage of *Gdf1/3-like*$^{-/-}$, the expression of *Brachyury* in notochord, *FoxQ2* in anterior ectoderm, *Hex* in anterior endoderm, *m-actin* in somites, *Otx* in forebrain and anterior pharyngeal endoderm, and *Wnt3* in hindbrain and spinal cord were absent (*Figure 2—figure supplement 3B*). Moreover, in a minority of *Gdf1/3-like*$^{+/-}$ embryos of the T0–T1 stage, the left-sided expression of *Pit* in preoral pit and *Pou4* in oral primordium disappeared, while the right-sided expression of *Nkx2.1* in endostyle and *FoxE4* in club-shaped gland became bilaterally symmetrical (*Figure 2C*). These data collectively indicated that *Gdf1/3-like* is indispensable for body axes formation in amphioxus and its loss-of-function leads to loss of dorsal, anterior, and left identities.

In vertebrates, transduction of Nodal signaling requires Gdf1/3 to form heterodimer with Nodal (*Montague and Schier, 2017*). To see whether Gdf1/3-like is similarly required for Nodal signaling transduction in amphioxus, we examined the expression of *Nodal* and a target gene (*Gsc*) of the Nodal signaling in *Gdf1/3-like*$^{-/-}$ embryos. Previous studies have used *Gsc* as a Nodal target in both vertebrates (*Thisse et al., 1994*; *Houston and Wylie, 2005*; *Bisgrove et al., 2017*) and invertebrates including amphioxus (*Onai et al., 2010*; *Saudemont et al., 2010*; *Morov et al., 2016*; *Zhang et al., 2019*). At the G1 stage, the expression of *Nodal* (most of them are probably maternal) was at a comparable level in *Gdf1/3-like*$^{+/+}$, *Gdf1/3-like*$^{+/-}$, and *Gdf1/3-like*$^{-/-}$ embryos (*Figure 2—figure supplement 4*). However, at the same stage the expression of *Gsc* was activated in the dorsal blastopore lip of *Gdf1/3-like*$^{+/+}$ embryos but disappeared or was reduced in *Gdf1/3-like*$^{-/-}$ embryos. Interestingly, we also noticed that in minority of *Gdf1/3-like*$^{+/-}$ embryos, the *Gsc* expression was reduced compared to *Gdf1/3-like*$^{+/+}$ embryos (*Figure 2—figure supplement 4*). These results suggested that in the absence of Gdf1/3-like, Nodal alone could not activate the Nodal signaling in amphioxus.

## Maternal and zygotic Nodal is necessary for amphioxus body axes formation

In amphioxus, *Nodal* is expressed both maternally and zygotically (*Onai et al., 2010*). To dissect its function during embryogenesis, we generated *Nodal* heterozygous animals and crossed them to analyze homozygous mutants. *Nodal*$^{-/-}$ embryos exhibited defects of L–R axis (*Figure 3—figure supplement 1*) as *Gdf1/3-like*$^{+/-}$ (shown above) and embryos in which late Nodal signaling activity was blocked (*Soukup et al., 2015*; *Li et al., 2017*). This result showed that zygotic Nodal is necessary for L–R patterning in amphioxus.

However, unlike *Gdf1/3-like*$^{-/-}$ mutants but like some *Gdf1/3-like*$^{+/-}$ mutants (*Figure 2A*), these *Nodal*$^{-/-}$ mutants do not show obvious dorsal–ventral or anterior–posterior defects (*Figure 3—figure supplement 1*). This is probably related to maternal *Nodal* expression. We therefore generated maternal *Nodal* mutants (M*Nodal*). Since zygotic *Nodal* mutants could not survive into adulthood, we screened for genetic mosaic females (founders) carrying oocytes of biallelic mutations (*Shi, 2022*) at the *Nodal* locus (*Figure 3—figure supplement 2A*). Two such founders (named founders 1 and 2) were identified from nearly one hundred of animals. *In situ* hybridization experiment revealed that around 50% and 20% of eggs released by the founders 1 and 2, respectively, showed no maternal *Nodal* mRNA accumulation (*Figure 3A, B*). In line with this, when eggs of the two founders were fertilized with WT sperms (*Figure 3—figure supplement 2B*), around 50% and 20% of them (M*Nodal*) showed a mild ventralized phenotype, respectively (*Figure 3C, D*, *Figure 3—figure supplement 2C*). Among embryos generated from the above two crosses, approximately 50% and 20% of them, respectively, showed reduced expression of anterior and dorsal markers (*Gsc*, *Chordin*, *Netrin*, *SoxB1a*, *Brachyury*, *Wnt3*, *m-actin*, and *FoxQ2*), and expanded expression of the ventral markers (*Evx* and *Ap2*) (*Figure 3E–F*, *Figure 3—figure supplement 3*). We also crossed the two founders with male *Nodal*$^{+/-}$ animals to generate maternal and zygotic *Nodal* mutants (MZ*Nodal*). MZ*Nodal* embryos, as expected, accounting for around 25% and 10% of offspring of the founders 1 and 2, respectively, displayed ventralized phenotypes being more severe than M*Nodal* mutants, but comparable to *Gdf1/3* mutants (*Figure 3C, D*, *Figure 3—figure supplement 2C*). Gene expression analysis showed that MZ*Nodal*

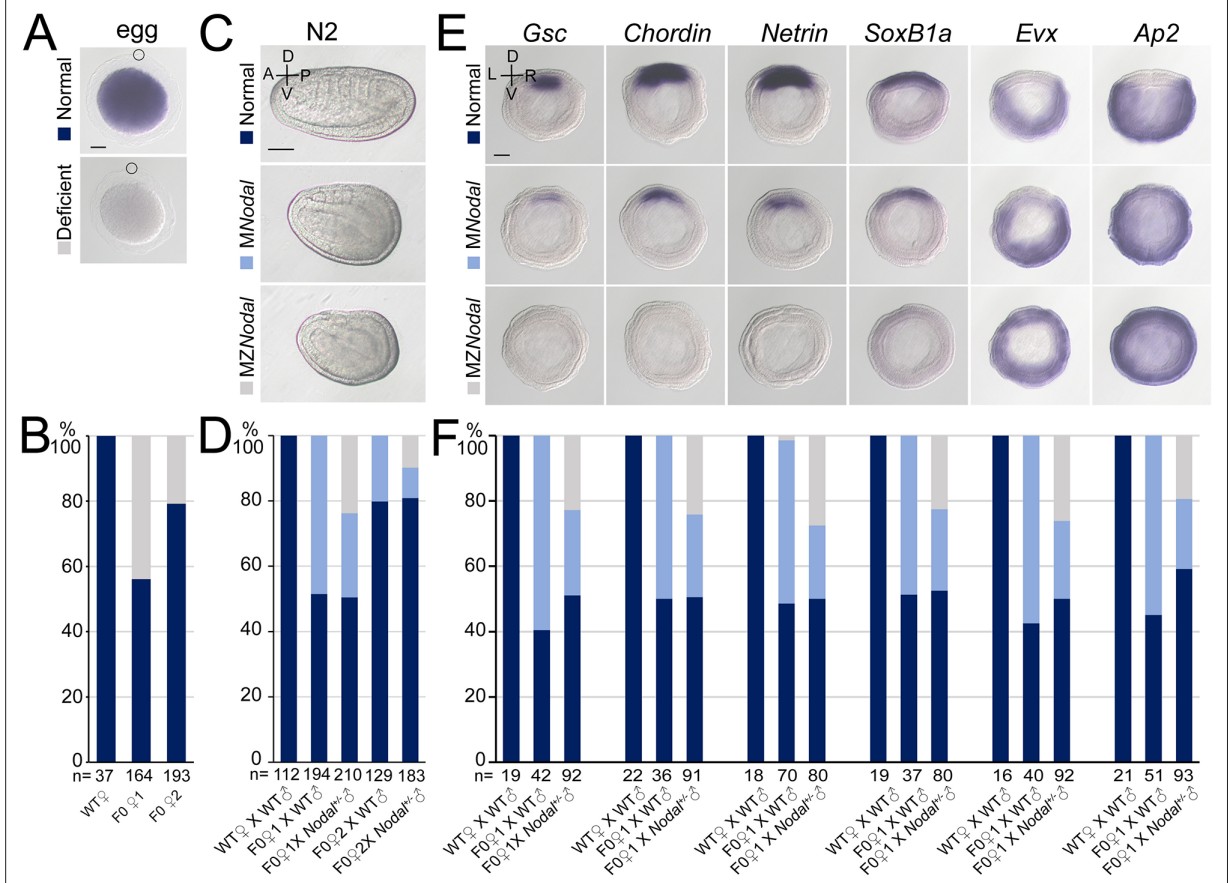

**Figure 3.** *Nodal* gene is required for amphioxus axis development. (**A**) *Nodal* expression in amphioxus eggs from different females (wild-type [WT], founder 1 [F0♀1] and founder 2 [F0♀2]). They were observed with animal pole to the top (circles indicate polar body). Two types of *Nodal* expression (normal and deficient) were observed. (**B**) Histogram showing the percentage of eggs of normal or deficient maternal *Nodal* accumulation from different females. (**C**) Phenotypic analyses of maternal (M*Nodal*) and maternal–zygotic (MZ*Nodal*) *Nodal* mutants. They were observed from the left side with anterior to the left at N2 stage (the A–P and D–V axes are labeled). Phenotype of the two mutants from a same pool at other stages are provided in *Figure 3—figure supplement 2*. (**D**) The percentage of embryos of different phenotypes as shown in (**C**). Embryos from five different crosses were examined. (**E**) The expression of marker genes in M*Nodal* and MZ*Nodal* mutants at G5 stage. All embryos were viewed from the blastopore with dorsal to the top (the D–V and L–R axes are labeled). (**F**) The percentage of embryos showing different expression patterns (normal, M*Nodal*, and MZ*Nodal*) as indicated in (**E**). Embryos from three different crosses were examined. Scale bars, 50 µm (**A, C, E**). The total number of analyzed eggs or embryos are listed under each column (**B, D, F**).

The online version of this article includes the following source data and figure supplement(s) for figure 3:

**Source data 1.** The number of eggs or embryos with different phenotypes or expression patterns.

**Figure supplement 1.** Phenotypic analysis of zygotic *Nodal* mutants and the expression of marker genes in zygotic *Nodal* mutants.

**Figure supplement 2.** Phenotypic analysis of maternal and maternal–zygotic *Nodal* mutants.

**Figure supplement 3.** The expression of marker genes in maternal and maternal–zygotic *Nodal* mutants at T0 stage.

**Figure supplement 4.** The expression of *Gsc* in maternal *Nodal* mutants injected with *Nodal* or *Gdf1/3-like* mRNA.

mutants lost most of the dorsal and anterior identities, as indicated by loss of expressions of the dorsal and anterior marker genes (*Gsc*, *Chordin*, *Netrin*, *SoxB1a*, *Brachyury*, *Wnt3*, *m-actin*, and *FoxQ2*), and expanded expressions of the ventral markers (*Evx* and *Ap2*) (*Figure 3E, F*, *Figure 3—figure supplement 3*). These results collectively indicated that maternal *Nodal*, probably zygotic *Nodal* as well, is required for regulating the D–V axis formation in amphioxus.

The phenotypes of MZ*Nodal* and *Nodal*^−/− are strikingly similar to those of *Gdf1/3-like*^−/− and *Gdf1/3-like*^+/−, respectively, suggesting that Nodal is indispensable for Gdf1/3-like activity during Nodal signal transduction. To further test this, we injected either *Nodal* or *Gdf1/3-like* mRNA into maternal M*Nodal* embryos and analyzed the *Gsc* expression at the G5 stage. We found that *Nodal*

mRNA injection could rescue *Gsc* expression in the M*Nodal* embryos, while *Gdf1/3-like* mRNA injection could not, although it expanded the *Gsc* expression domain in normal embryos (*Figure 3—figure supplement 4*). This result demonstrated that Gdf1/3-like alone (without Nodal) is unable to activate Nodal signaling in amphioxus.

### *Gdf1/3-like* hijacked the regulatory region of *Lefty*

As *Gdf1/3-like* is linked to *Lefty* in a head-to-head way in all sequenced amphioxus genomes and the two genes exhibit a similar expression pattern during embryogenesis, we hypothesized that the intergenic region between the two genes might include most if not all cis-regulatory elements (such as enhancers) required for their expression. To test this, we cloned the intergenic region (about 4 kb) into a pminiTol2 plasmid carrying a *mCherry* reporter and generated a stable amphioxus transgenic line carrying it (*Figure 4A*). Whole-mount *in situ* hybridization analysis revealed that the transcription of *mCherry* was similar (although not identical) to that of endogenous *Gdf1/3-like* and *Lefty* with simultaneous initiation and D–V and L–R asymmetric expression pattern (*Figure 4A*). We further made a dual-reporter pminiTol2 construct, in which the coding sequences of *eGFP* and *mCherry* were inserted into the two ends of the 4 kb region, respectively, and injected it into amphioxus embryos. *In situ* analysis of transient transgenic embryos showed that at the G1 stage, the 4 kb sequence could simultaneously drive *eGFP* and *mCherry* transcription in a similar pattern in the dorsal organizer region of the injected embryos (*Figure 4B*). These results indicated that the intergenic sequence indeed contains sequence elements required for regulating the expression of the two linked genes.

After demonstrating the bidirectional activity of the 4 kb intergenic sequence in driving *Lefty* and *Gdf1/3-like* genes, we then asked which one of the two genes originally used this region to regulate its expression. In sea urchin and vertebrate embryos, the expression pattern of *Lefty* and *Gdf1/3* genes are different, with the former being zygotically activated at blastula stage and unilaterally expressed from gastrula/neurula stage (*Meno et al., 1996*; *Meno et al., 1997*; *Bisgrove et al., 1999*; *Ishimaru et al., 2000*; *Tanegashima et al., 2000*; *Duboc et al., 2005*; *Duboc et al., 2008*), and the latter being maternally preloaded and bilaterally expressed from gastrula/neurula stage (*Supplementary file 1b*). Moreover, *Lefty* expression depends on Nodal signaling in these two groups (*Saudemont et al., 2010*; *Bisgrove et al., 2017*), while *Gdf1/3* expression does not, at least in sea urchin (*Range et al., 2007*). As demonstrated above, *Lefty* expression pattern in amphioxus is similar to its orthologs in sea urchin and vertebrates, but *Gdf1/3-like* expression follows essentially that of *Lefty*. Additionally, we and others also showed that *Lefty* expression in amphioxus embryos depends on Nodal signaling (*Soukup et al., 2015*; *Morov et al., 2016*; *Li et al., 2017*; *Zhang et al., 2019*). These results therefore implied that the 4 kb region was initially responsible for *Lefty* expression, and then hijacked by *Gdf1/3-like* after it was translocated to the current locus in amphioxus ancestor. To evaluate this scenario further, we examined *Gdf1/3-like* expression in the M*Nodal* embryos to see if it is dependent on Nodal signaling as that of *Lefty*. The result showed that, in M*Nodal* embryos, the expression of *Gdf1/3-like* and *Lefty* were both significantly reduced or abolished at the early gastrula stages (G1–G3), although their initial expressions were not affected at the blastula stage (256-cell) and their expressions re-appeared at late gastrula stage (G5) (*Figure 4—figure supplement 1*). To examine this more directly, we injected amphioxus embryos with the above dual-reporter construct and then treated them with Nodal signaling inhibitor SB505124 to see if the expression patterns of both reporter genes are affected. *In situ* result showed that compared to untreated embryos, SB505124-treated embryos exhibited decreased expression for both *eGFP* and *mCherry* genes (*Figure 4B*). This indicates that the 4 kb region includes the sequence elements required for *Lefty* and *Gdf1/3-like* expression regulation by Nodal signaling. Together, these results imply that the regulatory logic of *Lefty* expression is conserved in different deuterostomes and the regulatory elements used to regulate *Lefty* expression was hijacked by *Gdf1/3-like* after its translocation next to *Lefty* gene in amphioxus ancestor.

## Discussion

During evolution, morphological innovations are usually accompanied by changes of GRNs essential for developmental processes (*Peter and Davidson, 2011*). However, the GRNs underpinning a trait in a common ancestor could also diverge in descendant lineages even as the trait itself remains conserved, a phenomenon called developmental system drift (*True and Haag, 2001*; *Haag and True,*

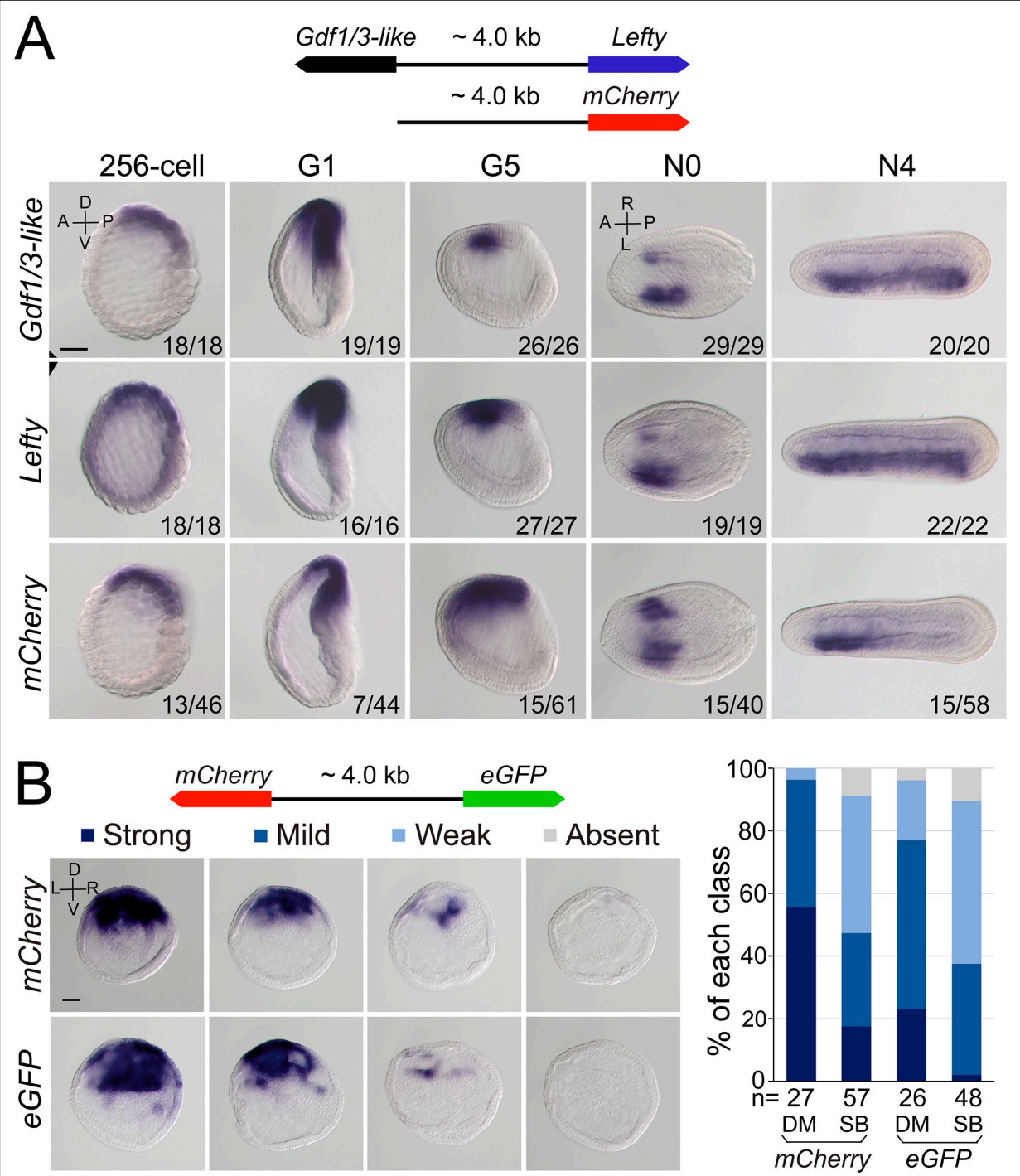

**Figure 4.** Regulatory activity of the intergenic region between *Gdf1/3-like* and *Lefty* genes. (**A**) The top schematic diagram shows the 4 kb region between *Gdf1/3-like* and *Lefty* and the construct used to generate stable amphioxus transgenic lines. Embryos used for *in situ* analysis were from a cross between a female founder and a wild-type (WT) male. The bottom panels show the expression patterns of *Gdf1/3-like*, *Lefty*, and *mCherry* reporter in transgenic embryos. Embryos at 256-cell to G5 stages were viewed from the left side (the A–P and D–V axes are labeled), and those at N0 and N4 stages were viewed from the dorsal side (the A–P and L–R axes are labeled). Numbers at the bottom right of each panel indicate the number of times the expression pattern was observed, out of the total number of examined embryos. (**B**) The left top schematic diagram shows the dual-reporter construct used in this analysis, and the panels below it shows the expression of *mCherry* and *eGFP* in embryos injected with the construct and then treated with dimethyl sulfoxide (DM) or SB505124 (SB). All embryos are at G1 stage viewed form the blastopore with dorsal to the top (the D–V and L–R axes are labeled). Four categories of expression (strong, mild, weak, and absent) for both *mCherry* and *eGFP* were observed in the dorsal blastopore lip of embryos, and their percentages are shown in the right histogram. The total number of analyzed embryos are listed under each column. Scale bars,

*Figure 4 continued on next page*

*Figure 4 continued*

50 µm. The effect of SB505124 treatment on Nodal signaling was validated by the phenotype of larvae treated (shown in *Figure 4—figure supplement 2*).

The online version of this article includes the following source data and figure supplement(s) for figure 4:

**Source data 1.** The number of embryos with different expression patterns.

**Figure supplement 1.** The expression of *Nodal*, *Gdf1/3-like*, and *Lefty* in maternal *Nodal* mutants at four different stages.

**Figure supplement 2.** The phenotype of larvae treated with SB505124.

*2021*). In either way, functional genetic evidence with a robust phylogenetic framework to demonstrate how GRN changes could have happened is sparse so far, although the underlying mechanism for the evolvability has been usually attributed to changes of cis-regulatory elements (*Peter and Davidson, 2011*). The most striking discovery in our study is that enhancer hijacking events could trigger the evolution of a GRN. In addition, through detailed analyses of available functional genetic

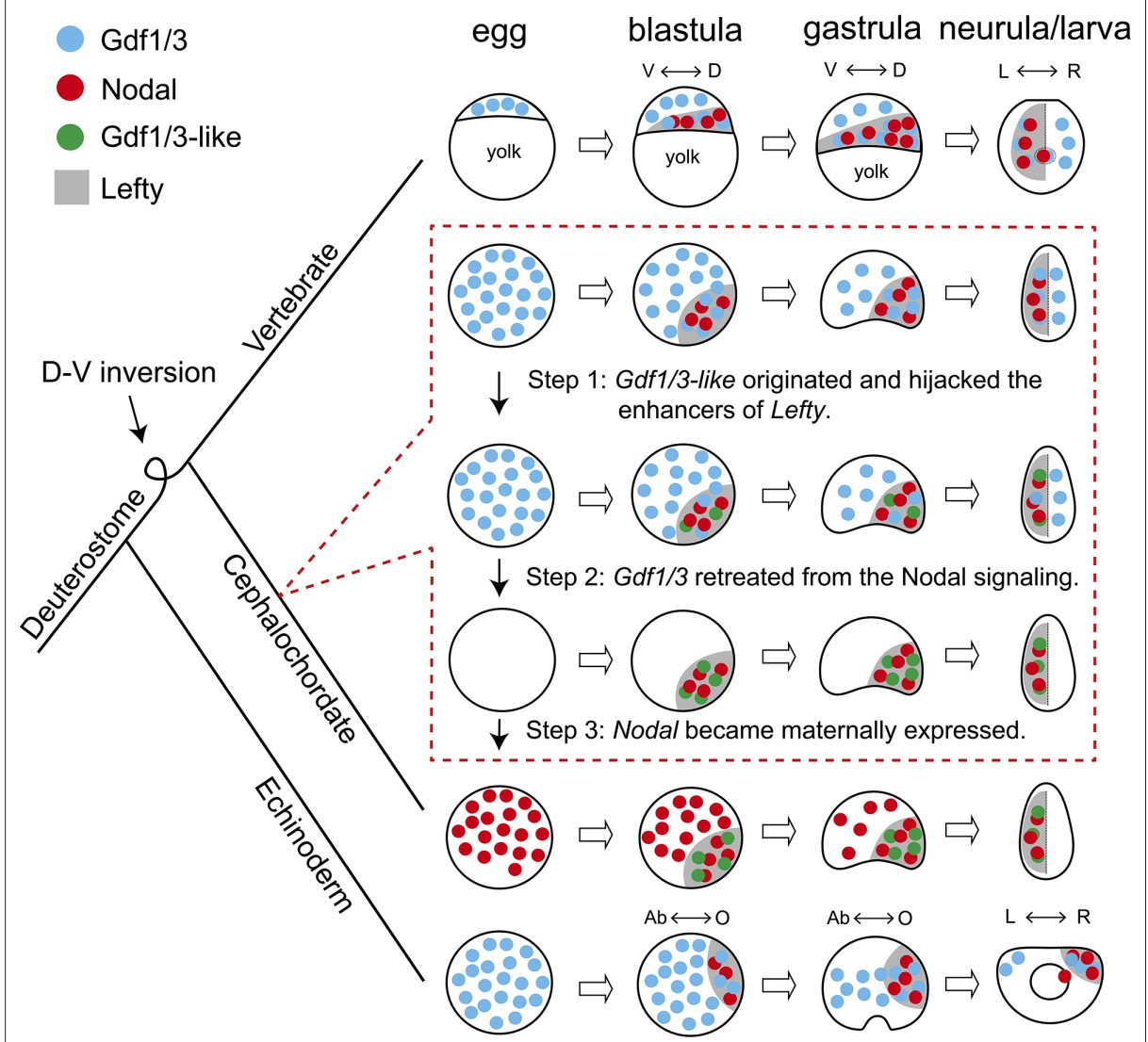

**Figure 5.** Scenario for evolution of the Nodal signaling in amphioxus. The situation in sea urchin (echinoderm) and zebrafish (vertebrate) represents an ancestral scenario which has been rewired and modified in amphioxus through at least three sequential steps as shown in dashed box. D–V, dorsal–ventral axis; L–R, left–right axis; Ab–O, oral–aboral axis. The dorsal–ventral orientation of Chordata was inversed relative to that of Ambulacraria during evolution (*Lowe et al., 2015*), thus the oral and left side of sea urchin corresponds to the dorsal and right side of zebrafish and amphioxus, respectively.

data below, we could demonstrate how a GRN could have evolved in a stepwise way, in the absence of 'molecular developmental fossils' (*Figure 5*).

Nodal signaling plays an essential role for patterning the D–V and L–R axis in deuterostomes (*Jia and Meng, 2021*). The expression patterns of the genes coding the ligands of the signaling pathway are conserved in extant echinoderms and vertebrates (*Supplementary file 1a, b*). Thus, the GRN of the Nodal signaling pathway revealed in these species most likely represents an ancestral scenario: Gdf1/3 is preloaded in an egg; by blastula stage, Nodal and Lefty appear zygotically; Nodal functions interdependently with the preexisting Gdf1/3 to activate the signaling pathway, while Lefty restrains the activity of the signaling pathway by inhibiting the signaling (*Figure 5*; *Müller et al., 2012*; *Montague and Schier, 2017*). Our data showed that this highly conserved GRN has been rewired and modified in cephalochordate amphioxus: Nodal is expressed maternally in an egg; ancestral Gdf1/3 lost its expression and function at embryonic stage, while *Gdf1/3-like* (a duplicate of *Gdf1/3*) recruited the regulatory elements of *Lefty* and is expressed in a similar pattern as *Lefty* from the blastula stage; Gdf1/3-like functions interdependently with the preexisting Nodal to activate the signaling pathway; and Lefty still acts as a feedback inhibitor of the signaling pathway (*Figure 5*). Our results and those from previous studies in other species enable us to infer that at least three sequential steps are probably required to evolve the GRN as found in extant amphioxus from an ancestral situation as reported in echinoderms and vertebrates (as discussed below): (1) the *Gdf1/3-like* originated and translocated adjacently to *Lefty* and hijacked its enhancers; (2) the *Gdf1/3* retreated from the Nodal signal; (3) the *Nodal* became maternally expressed (*Figure 5*).

Overexpression of *Gdf1/3* in zebrafish and *Xenopus* embryos have no effects on axes development (*Dale et al., 1993*; *Bisgrove et al., 2017*). It is therefore expected that the emergence of *Gdf1/3-like* would not affect the development and viability of the ancestor of extant amphioxus (i.e. step 1, *Figure 5*). In contrast, disruption of maternal *Gdf1/3* function is lethal in sea urchin and zebrafish embryos (*Range et al., 2007*; *Bisgrove et al., 2017*; *Montague and Schier, 2017*; *Pelliccia et al., 2017*). This indicates that the retreatment of the *Gdf1/3* ortholog from the Nodal signaling (step 2) could not happen before the emergence of *Gdf1/3-like* and its recruitment of the *Lefty* enhancers (step 1). Similarly, overexpression of Nodal in the presence of maternal Gdf1/3 led to defects in axis formation of zebrafish embryos (*Bisgrove et al., 2017*), suggesting that Nodal could not become a maternal factor before the retreatment of *Gdf1/3* (i.e. step 2 happened before step 3). These suggested that the 1–2–3 ordered events could have led to the situation in extant amphioxus (*Figure 5*). Injection of *Gdf1/3* mRNA to the yolk syncytial layer, where *Nodal* is expressed endogenously, was sufficient to rescue M*Gdf1/3* defects in zebrafish, thus the ubiquitous distribution of maternal *Gdf1/3*, in the region where *Nodal* is not expressed, is dispensable (*Montague and Schier, 2017*). Once the *Gdf1/3-like* started its current expression pattern by hijacking *Lefty* enhancers, the function of *Gdf1/3-like* and *Gdf1/3* became redundant, allowing the retreatment of *Gdf1/3* from the signaling pathway (i.e. step 2, *Figure 5*). As discussed in the study of zebrafish (*Montague and Schier, 2017*), preloading of a maternal factor (Gdf1/3 in zebrafish or Nodal in amphioxus) could be instrumental for ensuring Nodal signaling initiation in a rapid and temporally reliable manner. Likewise, the establishment of Nodal as a maternal factor may be a compensation mechanism for the retreatment of ubiquitously maternal Gdf1/3 (i.e. step 3, *Figure 5*). Zygotic *Nodal* mutants failed to form L–R axis but formed D–V axis normally in amphioxus (*Figure 3—figure supplement 1*), suggesting that maternal *Nodal* is redundant to compensate the D–V axis formation. However, maternal *Nodal* mutants formed only partial anterior and dorsal structures (*Figure 3*, *Figure 3—figure supplements 2 and 3*), suggesting that the expression of zygotic *Nodal* might be delayed in living amphioxus, compared to that of the amphioxus ancestor before step 3. Our proposed scenario highlights that enhancer hijacking by *Gdf1/3-like* was a triggering and probably 'neutral' step for the rewiring of the body axes GRN in amphioxus ancestors.

Although developmental system drift is probably universal in long evolutionary periods (*Palmer, 2004*), it has been rarely possible to analyze whether there is selective advantage of such drifts. Co-expressed gene pairs sharing intergenic enhancers have been reported in various cases (*Ueda et al., 2006*; *Sumiyama and Tanave, 2020*; *Zinani et al., 2021*; *Zinani et al., 2022*), and often fall into the same functional categories (*Zinani et al., 2022*). In zebrafish, the paired *her1* and *her7* can provide robustness for segmentation patterning (*Zinani et al., 2021*). The two proteins need to form dimers to inhibit their own transcription, thus forming a negative-feedback loop to maintain a stable level of both proteins in cells (*Zinani et al., 2021*). In our case, Gdf1/3-like functions as an activator of the

signaling that maintains the transcriptions of both *Gdf1/3-like* and *Lefty* in a positive-feedback loop, while Lefty acts as a repressor of the signaling forming a negative-feedback loop (*Li et al., 2017*; *Zhang et al., 2019*). The co-expression of *Gdf1/3-like* and *Lefty* gene pair (*Figure 1B*, *Figure 1— figure supplements 2–5*) achieved by sharing regulatory region (*Figure 4*) likely safeguards a dosage balance between the positive- and negative-feedback loop. This is likely an advantage for robust pattern formation as suggested in other gene pairing systems (*Zinani et al., 2022*).

Gdf1/3 is one of the reported examples of small-scale gene duplications in amphioxus (*Brasó-Vives et al., 2022*), anurans, and mammals (*Opazo and Zavala, 2018*). However, these genes appear to have evolved differently: in anurans and mammals both *Gdf1/3* paralogs retain the ancestral function in axis formation (*Birsoy et al., 2006*; *Andersson et al., 2007*; *Vonica and Brivanlou, 2007*), while in amphioxus only *Gdf1/3-like* undertakes this function but *Gdf1/3* seems to have lost its original function. As discussed above, hijacking of *Lefty* enhancers by *Gdf1/3-like* had enabled *Gdf1/3-like* to be functionally redundant to *Gdf1/3*, allowing *Gdf1/3* to retrieve from its ancestral function in regulating body axis formation through changing its expression pattern. In addition, the redundancy of *Gdf1/3-like* appears to have also relaxed the selection pressure on *Gdf1/3*-coding region, since *Gdf1/3-like* mRNA is more efficient in inducing anterior and dorsal expansion than that of *Gdf1/3* (*Figure 2—figure supplement 2*).

We have demonstrated that functional genetic data could be used to dissect the route of GRNs evolution, once there is available data from different species for comparisons. Our study represents an example that illustrated how the GRNs could have shifted in a stepwise way without altering conserved phenotypes, such as body axes that has persisted at least since the Early Cambrian period (*Martindale, 2005*; *Dunn et al., 2014*). Our case also showed that enhancer hijacking could be a mechanism underlying the evolution of developmental GRNs, in addition to the changes of cis-regulatory elements.

## Methods
### Animals and embryos
WT amphioxus (*B. floridae*) was obtained from Jr-Kai Yu's lab and bred in the aquaculture system as reported previously (*Li et al., 2012*). Mature individuals were induced to lay eggs and release sperm by thermal shock (*Li et al., 2013*). The developmental stages of amphioxus embryos were defined as recently described (*Carvalho et al., 2021*).

### Gene structures and synteny
Positions of *Lefty*, *Gdf1/3-like*, *Bmp2/4*, and *Gdf1/3* genes in the genomes of *B. floridae*, *B. belcheri*, *B. japonicum*, and *B. lanceolatum* were determined according to two recent studies (*Brasó-Vives et al., 2022*; *Huang et al., 2023*), while positions of these genes in sea urchin (*S. purpuratus*) were retrieved directly from Echinobase, in other species including zebrafish (*D. rerio*), elephant shark (*C. milii*), vase tunicate (*C. intestinalis*), acorn worm (*S. kowalevskii*), polychaete worm (*C. teleta*), and fruit fly (*D. melanogaster*) were retrieved directly from Ensembl database.

### Phylogenetic analyses
Amino acid sequences of Gdf1/3 and related TGF-β factors were aligned by ClustalX integrated in MEGA5 software (*Tamura et al., 2011*). The aligned sequences were then used to construct a phylogenetic tree using the maximum likelihood method with PhyML (http://atgc.lirmm.fr/phyml/; *Guindon et al., 2010*). The substitution model was automatically selected by Smart Model Selection in PhyML (*Lefort et al., 2017*).

### Quantitative real-time polymerase chain reaction
Embryos or eggs (about 200 per sample) were harvested at desired stages and used to extract total RNA by TRIzol reagent (Ambion). cDNA was synthesized using HiScript III RT SuperMix (+gDNA wiper) kit (Vazyme). Quantitative real-time polymerase chain reaction analysis was performed with ChamQ Universal SYBR qPCR Master Mix kit (Vazyme) on a CFX96 Touch Real-Time PCR Detection System (Bio-Rad) under the conditions of 95°C for 2 min, 40 cycles at 95°C for 5 s, 60°C for 30 s. Expression levels of *Nodal*, *Gdf1/3*, *Gdf1/3-like*, and *Lefty* were normalized to that of *Gapdh*

(glyceraldehyde-3-phosphate dehydrogenase). Graphs were finished with GraphPad Prism 9 software. Primers for quantitative real-time polymerase chain reaction analysis and their sequences were as follows: BfGdf1/3-like-RT-F (5′-CAAGGGCAAATATCACGACA-3′), Bf Gdf1/3-like-RT-R (5′-TTCACGTC GTCTCTGTCGAA-3′), BfNodal-RT-F (5′-GGACAGACCTCAACGTCACCC-3′), BfNodal-RT-R (5′-CTGA AGACACGCACGGAAAGT-3′), BfLefty-RT-F (5′-CACTGACGCCAGTGGTGCA-3′), BfLefty-RT-R (5′-CGTTGTTGAAAGACTTTCGAGT-3′), BfGdf1/3-RT-F (5′-TTCTCGGCTTTCGTGAACGG-3′), BfGdf1/3-RT-R (5′-ACAGTCCAACCATTTTCGGCA-3′), Gapdh-RT-F (5′-GGTGGAAAGGTCCTGCTCTC-3′), and Gapdh--RT-R (5′-CTGGATGAAAGGGTCGTTAATGG-3′).

## Overexpression experiments

Coding sequences of *Nodal* and *Gdf1/3-like* were amplified from cDNA templates of amphioxus embryos and ligated into the pXT7 vector using T4 DNA ligase (Promega). Due to low expression level, coding exons of *Gdf1/3* were individually amplified from genomic DNA templates and then assembled into the pXT7 vector using a Gibson cloning kit (New England Biolabs). *Nodal*, *Gdf1/3*, and *Gdf1/3-like* mRNA were synthesized using T7 mMESSAGE mMACHINE kit (Ambion). Unfertilized eggs were injected with mRNAs and fertilized as previously described (*Liu et al., 2013*). The injected embryos were treated, with 0.2% dimethyl sulfoxide (as control) or 50 μM SB505124, from 4-cell stage to G1 stage and were fixed at G4 or N3 stage for whole-mount *in situ* hybridization.

## Generation of mutant animals and embryos

*Gdf1/3* mutant were generated using CRISPR/Cas9 system targeting a site (*Gdf1/3-sg*RNA: 5′-GGCC CGCTGTAGCGATGA-3′) in the first exon. The process of generating founders of *Gdf1/3* gene knock-out was implemented as our previous study (*Su et al., 2020*). F1 *Gdf1/3*$^{+/-}$ carrying −25 bp mutation were intercrossed to generate *Gdf1/3*$^{-/-}$ (*Figure 2—figure supplement 1*). A pair of primer (*Gdf1/3*-Cas9-F1: 5′-TACCACACATCACCCGGACT-3′/*Gdf1/3*-PCR-R1: 5′-CACATCCTCGTCTTCC GGTC-3′) and *Sfc*I enzyme were used for genotyping and mutation type analysis.

*Gdf1/3-like* mutants were generated with a TALEN pair (*Gdf1/3-like*-Fw: 5′-TTCGACAGAGAC GAC-3′/*Gdf1/3-like*-Rv: 5′-TGCACGGCGCTCACGA-3′) targeting the coding region of the second exon as previously reported (*Li et al., 2014*; *Li et al., 2017*). F1 heterozygotes carrying a one base pair insertion (+1 bp) were used and intercrossed to generate homozygote mutants (*Gdf1/3-like*$^{-/-}$) (*Figure 2—figure supplement 3A*). Embryos or tiny tips of adult tail were lysed with Animal Tissue Direct PCR Kit (Foregene) to release the genomic DNA. Genomic region spanning the target site were amplified with primer *Gdf1/3-like*-TALEN-F: 5′-CGTGACGTACTCCGTGTCTG-3′/*Gdf1/3-like*-TALEN-R1: 5′-GCTGAAGTGTGGGCAAGAGT-3′ or *Gdf1/3-like*-TALEN-R0: 5′-CCGTTTGCAGATGTTG CCG-3′. The amplicons were digested with *Stu*I to test the mutations and sequenced to recognize the mutation types.

*Nodal* heterozygotes (*Nodal*$^{+/-}$) were generated using a TALEN pair (Nodal-Fw2/Rv2) in previous study (*Shi et al., 2020*). Male and female *Nodal*$^{+/-}$ carrying identical mutation (-7 bp) were intercrossed to generate *Nodal*$^{-/-}$ (*Figure 3—figure supplement 1A*). Genotyping for *Nodal* mutants were implemented as previous study (*Shi et al., 2020*). A TALEN pair (Nodal-Fw2/Rv2) assembled previously (*Shi et al., 2020*) and a sgRNA (*Nodal*-sgRNA:5′-GGCGGAGAGGGTCTGACGCT-3′) reported previously (*Su et al., 2020*) were used to generate hundreds of chimeric female founders. By adulthood stage, among nearly 100 of animals, an individual generated with Nodal-Fw2/Rv2 (named founder 1, F0♀1) and another individual generated with *Nodal*-sgRNA (named founder 2, F0♀2) laid eggs lacking maternal *Nodal* accumulation (*Figure 3—figure supplement 2A*) with 50% and 20% ratio, respectively. Each female founder was crossed with a male WT, respectively, to generated maternal *Nodal* mutants (M*Nodal*) and crossed with a male *Nodal*$^{+/-}$, respectively, to generated maternal–zygotic *Nodal* mutants (MZ*Nodal*) (*Figure 3—figure supplement 2B*).

## Generation of transgenic animals and embryos

The intergenic region between *Gdf1/3-like* and *Lefty* was cloned into the pmini-Chordin-mCherry (*Shi et al., 2018*) by replacing its *Chordin* promoter with the intergenic region. The generated pmini-Lefty-mCherry construct was used to generate *Lefty::mCherry* transgenic amphioxus with method reported previously (*Shi et al., 2018*). The intergenic region, the coding sequence of *mCherry* and *eGFP* sequence were linked into the pminiTol2 plasmid with a Gibson cloning kit (New England Biolabs)

to generate pmini-eGFP-Lefty-Gdf1/3-like-mCherry construct. The later was injected into amphioxus embryos. The injected embryos were treated, with 0.2% dimethyl sulfoxide (as control) or 50 µM SB505124, from 4-cell stage to G1 stage.

## Whole-mount *in situ* hybridization and imaging

The RNA probes used in present study were prepared previously (*Li et al., 2017*; *Shi et al., 2018*; *Zhang et al., 2019*), except the probe of *eGFP* and *Gdf1/3* gene, which was prepared using the same method as described in a previous study (*Li et al., 2017*). Embryos at desired developmental stages were fixed with 4% paraformaldehyde in MOPS buffer (PFA–MOPS, wt/vol) and then stored in 80% ethanol in $H_2O$ (vol/vol) at −20°C until needed. Whole-mount *in situ* hybridization was performed as previously described (*Yu and Holland, 2009*). After staining, the embryos were mounted in 80% glycerol (vol/vol) for photographing under an inverted microscope (Olympus IX71). Living amphioxus embryos or larvae were observed using the same microscope. We use Photoshop software to automatically stitch together photos of different parts of the embryo or larva, when a single field of view cannot display the whole mount of embryo or larva.

The double *in situ* hybridization of *Lefty* and *Gdf1/3-like* was achieved by using amplification-based single-molecule *in situ* hybridization (asmFISH) method as described in a previous study (*Lin et al., 2021*), which hereby was performed using SEERNA ISH RNA Fluorescence *in Situ* Detection Kit (SEERNA Bioscience, Xiamen, China, Cat# SP1001, SP1002, and SP1003) under the guidance of standard protocol of the manual. The details are as follows. The label probes of *Lefty* and *Gdf1/3-like* were conjugated to Cy5 and Alexa fluor 488, respectively (provided in the Kit). For asmFISH, embryos at desired developmental stages were fixed in 4% PFA–MOPS (wt/vol) and then stored in methanol at −20°C until needed. Before hybridization, the embryos were permeabilized in 0.1 M HCl containing 0.1 mg/ml pepsin (Sigma, Cat# P7012) at 37°C (for 5 min at 256-cell and G1 stage, 9 min at G4 stage, 30 min at N2 stage) and washed with Wash Buffer (provided in the Kit). The embryos were then incubated in order with the following solutions: target probes in hybridization mix at 46°C for 4 hr, ligation mix at 37°C for 30 min, splint primers in the circularization mix at 37°C for 30 min, amplification mix for rolling circle amplification at 30°C overnight, label probes in hybridization buffer at room temperature for 30 min. Embryos were washed with the Wash Buffer three times after each step. Finally, the embryos were placed in a Mounting medium (Solarbio, Cat# S2100) containing 0.5 µg/ml 4,6-diamidino-2-phenylindole (Sigma, Cat# D8417), which were then ready for image acquisition using the Zeiss LSM980 microscope.

## Acknowledgements

We thank all team members of the GL and QQ laboratories, and Prof. Zhenghong Zuo for helpful discussions. This work is supported by grants from the National Natural Science Foundation of China (nos. 32070458, 31872186, 32070815, and 32061160471), the Youth Innovation Fund Project of Xiamen (3502Z20206032) and the start-up fund of Xiamen University.

## Additional information

### Funding

| Funder | Grant reference number | Author |
| --- | --- | --- |
| National Natural Science Foundation of China | 32070458 | Guang Li |
| National Natural Science Foundation of China | 31872186 | Guang Li |
| National Natural Science Foundation of China | 32070815 | Yiquan Wang |
| National Natural Science Foundation of China | 32061160471 | Guang Li |

| Funder | Grant reference number | Author |
|--------|------------------------|--------|
| Youth Innovation Foundation of Xiamen | 3502Z20206032 | Guang Li |
| Xiamen University | start-up fund of Xiamen University | Qingming Qu |

The funders had no role in study design, data collection, and interpretation, or the decision to submit the work for publication.

### Author contributions

Chenggang Shi, Conceptualization, Data curation, Formal analysis, Validation, Investigation, Visualization, Methodology, Writing - original draft, Writing - review and editing; Shuang Chen, Huimin Liu, Rongrong Pan, Shiqi Li, Yanhui Wang, Xiaotong Wu, Jingjing Li, Xuewen Li, Chaofan Xing, Xian Liu, Investigation; Yiquan Wang, Conceptualization, Funding acquisition, Project administration; Qingming Qu, Conceptualization, Writing - original draft, Writing - review and editing; Guang Li, Conceptualization, Supervision, Funding acquisition, Investigation, Writing - original draft, Project administration, Writing - review and editing

### Author ORCIDs

Chenggang Shi (ID) http://orcid.org/0000-0002-5592-2761
Qingming Qu (ID) http://orcid.org/0000-0002-8291-8493
Guang Li (ID) http://orcid.org/0000-0002-5543-5349

### Decision letter and Author response

Decision letter https://doi.org/10.7554/eLife.89615.sa1
Author response https://doi.org/10.7554/eLife.89615.sa2

## Additional files

### Supplementary files

• Supplementary file 1. The expression pattern of Nodal and Gdf1/3 gene in representative deuterostomes studied previously. (**a**) The expression pattern of *Nodal* gene in representative deuterostomes; (**b**) the expression pattern of *Gdf1/3* and its paralogs in representative deuterostomes.

• MDAR checklist

### Data availability

All data generated or analysed during this study are included in the manuscript and supporting file. Source data files containing the numerical data used to generate the figures have been provided.

The following previously published datasets were used:

| Author(s) | Year | Dataset title | Dataset URL | Database and Identifier |
|-----------|------|---------------|-------------|-------------------------|
| Marlétaz F, Firbas PN, Maeso I, Tena JJ | 2018 | Functional genomic and transcriptomic analysis of amphioxus and the origin of vertebrate genomic traits [RNA-Seq] | https://www.ncbi.nlm.nih.gov/geo/query/acc.cgi?acc=GSE106430 | NCBI Gene Expression Omnibus, GSE106430 |
| Huang Z | 2020 | Branchiostoma floridae x Branchiostoma belcheri isolate: bbbf Genome sequencing and assembly | https://www.ncbi.nlm.nih.gov/bioproject/PRJNA603158 | NCBI BioProject, PRJNA603158 |
| Huang Z | 2020 | Branchiostoma floridae x Branchiostoma belcheri isolate: bbbf Genome sequencing and assembly | https://www.ncbi.nlm.nih.gov/sra/PRJNA603159 | NCBI Sequence Read Archive, PRJNA603159 |

*Continued on next page*

*Continued*

| Author(s) | Year | Dataset title | Dataset URL | Database and Identifier |
|---|---|---|---|---|
| Huang Z | 2020 | Branchiostoma floridae x Branchiostoma japonicum | https://www.ncbi.nlm.nih.gov/genome/?term=PRJNA647830 | NCBI Taxonomy, PRJNA647830 |
| Huang Z | 2020 | Genome sequencing and assembly of three amphioxuses | https://www.ncbi.nlm.nih.gov/bioproject/PRJNA602496 | NCBI BioProject, PRJNA602496 |
| Brasó-Vives M, Marlétaz F, Echchiki A, Mantica F, Acemel RD, Gómez-Skarmeta JL | 2022 | Branchiosotma lanceolatum reference genome and gene annotation | https://www.ebi.ac.uk/ena/browser/view/PRJEB49647 | European Nucleotide Archive, PRJEB49647 |

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
