## [Editor Report]

This valuable work advances our understanding of how deuterostome body plan is established during development as well as how the gene regulatory network governing early embryogenesis has been rewired during animal evolution. The evidence supporting the conclusions is compelling, with a suite of molecular-level experiments. The work will be of broad interest to developmental biologists.

---

## [Decision Letter]

**Decision letter after peer review:**

Thank you for submitting your work entitled "Evolution of the gene regulatory network of body axis by enhancer hijacking in amphioxus" for consideration by *eLife*. Your article has been reviewed by 3 peer reviewers, one of whom is a member of our Board of Reviewing Editors, and the evaluation has been overseen by Claude Desplan as the Senior Editor.

We congratulate you on the valuable findings reported in the manuscript. However, although most of the supporting arguments are compelling, some are incomplete and you still need to provide arguments to convince the reviewers with 1) molecular phylogeny of Gdf1/3 in relation to Bmp2/4 and 2) co-expression of Gdf1/3 and Lefty. For the latter, adding double in situ hybridization data to the manuscript seems a straightforward solution. Please also respond to all the points raised below by the reviewers by incorporating the suggestions in the manuscript or arguing why they might not be necessary.

*Reviewer #1 (Recommendations for the authors):*

This manuscript focuses on amphioxus development and reports remarkable changes in the role of the previously documented regulatory gene, Gdf1/3-like, a duplicate gene unique to the amphioxus that replaces that of Gdf1/3 and Nodal genes. Their functional assays supported by a suite of molecular-level experiments provides solid evidence of an evolutionary rewiring of gene regulatory network governing early embryogenesis to confer a new role that resembles the Lefty gene on the Gdf1/3-like gene.

The authors of this manuscript performed gene expression profiling of the two Gdf1/3 orthologs in amphioxus and analyzed loss-of-function phenotypes of the Gdf1/3-like gene as well as Nodal mutants. Their cross-species comparisons suggested the overwriting of the embryonic regulatory processes in amphioxus – the additional Gdf1/3 gene retreated from the Nodal signalling and acquired the maternal expression originally exhibited by the Lefty gene. The study provides a novel insight into how gene duplication conferred novel embryonic regulatory systems. The study is well designed and soundly discussed and concluded, but the authors are suggested to improve the manuscript by considering the points listed below.

1. The naming and duality of mammalian Gdf1/3 orthologs are not mentioned in the Introduction. When first introduced, they should appear as separate genes Gdf1 and Gdf3 at least once. This is confusing to readers who have not dealt with those genes.

2. Line 27 in the Abstract – The phrase 'share regulatory region' should have an article or should be in plural, depending on the context.

3. Line 231 'the GRN could also evolve or diverge' – what does this mean exactly? This part should be clarified.

4. Line 298 'Amphioxus has been shown to possess small-scale gene duplications' – Small-scale gene duplications should have occurred and be observed in any large genomes. This part of the sentence should be deleted. The next sentence can start with something like 'Gdf1/3 is one of the reported examples of small-scale gene duplications'.

5. How distant is Gdf1/3 from Bmp2/4 in the different deuterostome species? This information should be reflected in Figure 1a or in the text. Small distance should make the gene linkage look more likely to have been generated through tandem gene duplication.

6. Line 576 'elephant Shark' -> elephant shark or elephant fish (if you respect how the species is called near its habitats, Australia and New Zealand).

7. The taxonomic presentation of the different lineages is rarely correct in the figures. Vertebrata, Echinodermata, Cephalochordata, … If it denotes the species therein, the words should be in plural like Vertebrates, Echinoderms, and Cephalochordates, if I understand correctly.

8. The middle part of the Abstract that is supposed to be describing the solid results of the study looks like a Discussion including interpretations and speculations (e.g., 'hijacking') rather than descriptions of approaches taken and the obtained data. The authors need to revise the second half of the Abstract by making a distinction in the tone between fact description and interpretation/speculation.

*Reviewer #2 (Recommendations for the authors):*

In this paper Shi et al. study the role of Gdf1/3 and its orthologue Gdf1/3-like during amphioxus embryonic development. The work uses KO animals for Gdf1/3, Gdf1/3-like and Nodal, and establishes that the conserved role of this gene in the control of embryonic axes is developed by the Gdf1/3-like gene while Gdf1/3 has lost this function. Finally, through the use of a transgenic animal, they show that the regulation of the expression of Gdf1/3-like is carried out through elements located in the region between this gene and Lefty, suggesting that after the duplication of Gdf, one of the duplicated genes has co-opted the regulatory elements of Lefty to carry out its function. Finally, an evolutionary scenario in which the expression changes of these genes and Nodal between amphioxus and other deuterostomes, as well as the acquisition of the Lefty-like Gdf1/3-like expression pattern after hijacking the regulatory elements, is proposed at the end of the paper.

I consider this work interesting and well developed. The question of the evolution of GRNs is important in EvoDevo, and this example is interesting as it shows that a GRN can be changed in order to maintain the same body plan. The experimental approaches are of high level, and the use of maternal mutants is, to my knowledge, the first example of this kind of approach using amphioxus.

*Reviewer #3 (Recommendations for the authors):*

The authors address the gene regulatory network of Nodal signalling in Amphioxus and how it has drifted compared to deuterostomes. By analysing the genomic structure, expression and specific mutants of the paralogue genes encoding the TGFb ligand Gdf1/3, a partner of Nodal, they convincingly demonstrate that Gdf1/3-like has replaced Gdf1/3 in Amphioxus for mediating Nodal signaling during axis determination. Mutant and rescue experiments show that Nodal and Gdf1/3-like are required together. Using complex and rigorous crossing strategies, they elegantly uncover a maternal contribution of Nodal to antero-posterior and dorso-ventral axis determination, acquired in Amphioxus. Based on close genomic proximity and expression patterns, they explore the hypothesis that Gdf1/3-like has hijacked Lefty enhancers to support this developmental system drift. Using transgenic lines, they provide evidence that the intergenic region between Gdf1/3-like and Lefty is sufficient to drive the specific expression pattern, and that it is sensitive to a drug targeting Nodal and TGFb signaling.

The genetic investigation is solid, including the generation of 3 mutant lines, rescue experiments, drug treatments and the generation of 2 transgenic lines, convincingly supporting the claim of gene regulatory network evolution. However, the main claim of enhancer hijacking is not completely established and will require strengthening or modulation.

Overall, this carefully performed study provides an example of a developmental system drift, which will be of broad interest to evolutionary and developmental biologists.

– Validation of Gdf1/3-like mutants should be supported by control experiments showing that Gdf1/3-like expression is impaired. Since TALEN can have off-targets, it is important to show that the associated pathway and neighbouring loci are not affected (Gdf1/3, Bmp2/4, Lefty, Nodal). The conclusion that "the expression of Gsc was already activated in Gdf1/3-like+/+ and most Gdf1/3-like+/- but not in Gdf1/3-like-/- embryos" is difficult to follow in Supplementary Figure 5 and should be quantified by qPCR.

– Similarly, zygotic Nodal mutants are not validated for impairment of Nodal signaling.

– The hypothesis that Gdf1/3-like has hijacked Lefty enhancers is relevant but requires strengthening. An alternative hypothesis could be that Gdf1/3-like duplication was translocated close to Lefty with its own enhancers.

Co-expression of Gdf1/3-like and Lefty is lacking, either by double ISH such as recent HCR development (PMID: 32394382) or from available single cell RNAseq dataset. Can putative common enhancer or factor binding sites be identified by analysis of the intergenic sequence? Is there any Foxh1, Smad-responsive motifs? Are both Gdf1/3-like and Lefty expressions affected upon mutation of the intergenic sequence?

Please comment on differences in Figure 4 between mcherry and endogenous gene expression, and differences between mcherry and egfp. Illustration of SB505124 treatment is lacking as well as control of the effect on Nodal signaling.

It is intriguing to see that in maternal Nodal mutant, Gdf1/3-like and Lefty expressions re-appears before zygotic Nodal. Does it imply that they have alternative inducers? This is also consistent with the partial effect of SB505124.

---

## [Author Response]

Essential revisionsReviewer #1 (Recommendations for the authors):This manuscript focuses on amphioxus development and reports remarkable changes in the role of the previously documented regulatory gene, Gdf1/3-like, a duplicate gene unique to the amphioxus that replaces that of Gdf1/3 and Nodal genes. Their functional assays supported by a suite of molecular-level experiments provides solid evidence of an evolutionary rewiring of gene regulatory network governing early embryogenesis to confer a new role that resembles the Lefty gene on the Gdf1/3-like gene.The authors of this manuscript performed gene expression profiling of the two Gdf1/3 orthologs in amphioxus and analyzed loss-of-function phenotypes of the Gdf1/3-like gene as well as Nodal mutants. Their cross-species comparisons suggested the overwriting of the embryonic regulatory processes in amphioxus – the additional Gdf1/3 gene retreated from the Nodal signalling and acquired the maternal expression originally exhibited by the Lefty gene. The study provides a novel insight into how gene duplication conferred novel embryonic regulatory systems. The study is well designed and soundly discussed and concluded, but the authors are suggested to improve the manuscript by considering the points listed below.1. The naming and duality of mammalian Gdf1/3 orthologs are not mentioned in the Introduction. When first introduced, they should appear as separate genes Gdf1 and Gdf3 at least once. This is confusing to readers who have not dealt with those genes.

Thanks for your suggestion. We added a sentence “Different from most deuterostomes which has one *Gdf1/3* gene, mammals and frogs have two such genes (namely *Gdf1* and *Gdf3*), derived from lineage-specific duplications (*Opazo and Zavala, 2018; Opazo et al., 2019*). For simplicity, we collectively call them *Gdf1/3* genes in this study.” in the introduction to clarify this. Additional information can be found in supplementary table 2.

2. Line 27 in the Abstract – The phrase 'share regulatory region' should have an article or should be in plural, depending on the context.

Thanks, we have corrected it in the revised manuscript.

3. Line 231 'the GRN could also evolve or diverge' – what does this mean exactly? This part should be clarified.

Thanks for pointing this out. We revised it to “the GRNs underpinning a trait in a common ancestor could also diverge in descendant lineages even as the trait itself remains conserved, a phenomenon called developmental system drift.”

4. Line 298 'Amphioxus has been shown to possess small-scale gene duplications' – Small-scale gene duplications should have occurred and be observed in any large genomes. This part of the sentence should be deleted. The next sentence can start with something like 'Gdf1/3 is one of the reported examples of small-scale gene duplications'.

Thanks, we revised the sentence according to your suggestion.

5. How distant is Gdf1/3 from Bmp2/4 in the different deuterostome species? This information should be reflected in Figure 1a or in the text. Small distance should make the gene linkage look more likely to have been generated through tandem gene duplication.

We have added the distance between *Gdf1/3* and *Bmp2/4* in Figure 1A in revised version.

6. Line 576 'elephant Shark' -> elephant shark or elephant fish (if you respect how the species is called near its habitats, Australia and New Zealand).

We have corrected it in our manuscript.

7. The taxonomic presentation of the different lineages is rarely correct in the figures. Vertebrata, Echinodermata, Cephalochordata, … If it denotes the species therein, the words should be in plural like Vertebrates, Echinoderms, and Cephalochordates, if I understand correctly.

Thanks for pointing this out. We have revised Figure 1 accordingly.

8. The middle part of the Abstract that is supposed to be describing the solid results of the study looks like a Discussion including interpretations and speculations (e.g., 'hijacking') rather than descriptions of approaches taken and the obtained data. The authors need to revise the second half of the Abstract by making a distinction in the tone between fact description and interpretation/speculation.

We have revised the Abstract according to the suggestion.

Reviewer #3 (Recommendations for the authors):The authors address the gene regulatory network of Nodal signalling in Amphioxus and how it has drifted compared to deuterostomes. By analysing the genomic structure, expression and specific mutants of the paralogue genes encoding the TGFb ligand Gdf1/3, a partner of Nodal, they convincingly demonstrate that Gdf1/3-like has replaced Gdf1/3 in Amphioxus for mediating Nodal signaling during axis determination. Mutant and rescue experiments show that Nodal and Gdf1/3-like are required together. Using complex and rigorous crossing strategies, they elegantly uncover a maternal contribution of Nodal to antero-posterior and dorso-ventral axis determination, acquired in Amphioxus. Based on close genomic proximity and expression patterns, they explore the hypothesis that Gdf1/3-like has hijacked Lefty enhancers to support this developmental system drift. Using transgenic lines, they provide evidence that the intergenic region between Gdf1/3-like and Lefty is sufficient to drive the specific expression pattern, and that it is sensitive to a drug targeting Nodal and TGFb signaling.The genetic investigation is solid, including the generation of 3 mutant lines, rescue experiments, drug treatments and the generation of 2 transgenic lines, convincingly supporting the claim of gene regulatory network evolution. However, the main claim of enhancer hijacking is not completely established and will require strengthening or modulation.Overall, this carefully performed study provides an example of a developmental system drift, which will be of broad interest to evolutionary and developmental biologists.– Validation of Gdf1/3-like mutants should be supported by control experiments showing that Gdf1/3-like expression is impaired. Since TALEN can have off-targets, it is important to show that the associated pathway and neighbouring loci are not affected (Gdf1/3, Bmp2/4, Lefty, Nodal).

We understand your concern, and thanks for your suggestion. However, it is notable that there is only 1-bp insertion (within in the 2^nd^ exon), but not whole locus deletion, in the *Gdf1/3-like* mutants we generated. Therefore, transcription of *Gdf1/3-like* would not be altered by the mutation itself and evaluating its expression cannot validate the mutants.

We validated the specificity of the *Gdf1/3-like* TALENs from the following aspects: firstly, as you suggested we aligned the coding sequence of *Gdf1/3-like* and those of *Gdf1/3*, *Bmp2/4*, *Lefty* and *Nodal*, and found that the *Gdf1/3-like* TALEN target site is not conserved in other genes; secondly, using the *Gdf1/3-like* TALEN target site as query, we BLAST-searched the *Branchiostoma floridae* genome (*Putnam et al., 2008*) in UCSC database (http://genome.ucsc.edu/cgi-bin/hgBlat) and found no extra sequences similar to the *Gdf1/3-like* target; thirdly, as we showed in the manuscript, knock-out of *Gdf1/3*, the closest relative of *Gdf1/3-like*, caused no body axis defects as *Gdf1/3-like* mutation. Moreover, previous studies showed that both inhibiting BMP activity (*Kozmikova et al., 2013*) and knocking-out *Lefty* gene (*Zhang et al., 2019*) in amphioxus caused phenotypes opposite to that observed in *Gdf1/3-like* mutants. All these results strongly suggested that the phenotype of *Gdf1/3-like* mutants were not resulted from off-target effects.

The conclusion that "the expression of Gsc was already activated in Gdf1/3-like+/+ and most Gdf1/3-like+/- but not in Gdf1/3-like-/- embryos" is difficult to follow in Supplementary Figure 5 and should be quantified by qPCR.

We have rephrased the sentence as " the expression of *Gsc* was activated in the dorsal blastopore lip of *Gdf1/3-like^+/+^* embryos but disappeared or was reduced in *Gdf1/3-like^-/-^* embryos. Interestingly, we also noticed that in a minority of *Gdf1/3-like^+/-^* embryos, the *Gsc* expression was reduced compared to *Gdf1/3-like^+/+^* embryos.". We also indicated the domain (the dorsal blastopore lip) of *Gsc* expression in each embryo by arrows in the revised figure (Figure 2—figure supplement 4) to make it easy to follow.

We would like to perform qPCR experiment to quantify the expression of *Gsc* in embryos with different genotypes. But such experiment is almost impossible to do, since at this stage *Gdf1/3-like^-/-^* embryos show no morphological difference from *Gdf1/3-like^+/+^* and *Gdf1/3-like^+/-^* ones.

– Similarly, zygotic Nodal mutants are not validated for impairment of Nodal signaling.

We have explained this concern above (point 1 of reviewer #3). Moreover, we analyzed the expression of *Nodal*, *Gdf1/3-like*, *Lefty* and *Pitx* in zygotic *Nodal* mutants and found that the expression of these genes were all abolished in the mutants (Author response image 1), as that observed in previous studies when Nodal signaling was inhibited (*Soukup et al., 2015; Li et al., 2017*).

**Author response image 1. sa2fig1:** 

– The hypothesis that Gdf1/3-like has hijacked Lefty enhancers is relevant but requires strengthening. An alternative hypothesis could be that Gdf1/3-like duplication was translocated close to Lefty with its own enhancers. Co-expression of Gdf1/3-like and Lefty is lacking, either by double ISH such as recent HCR development (PMID: 32394382) or from available single cell RNAseq dataset.

As we have already stated in the ms, we raised the hypothesis that *Gdf1/3-like* hijacked the regulatory region of *Lefty* mainly based on 1) *Gdf1/3-like* is expressed like *Lefty* of amphioxus, but not *Gdf1/3* of other deuterostomes. 2) Like *Lefty*, *Gdf1/3-like* expression is regulated by Nodal signaling. However, *Gdf1/3* expression in other deuterostomes is independent of Nodal signaling. 3) *Gdf1/3-like* and *Lefty* are linked in a head-to-head fashion, and the intergenic sequence between them can simultaneously drive dual reporter genes expression as those of endogenous *Gdf1/3-like* and *Lefty*. 4). No metazoan is known to have a *Gdf1/3* related gene linked to *Lefty* other than amphioxus, suggesting that *Gdf1/3-like* did not originate together with *Lefty*, as this scenario would require independent losses of *Gdf1/3-like* in numerous lineages. We think these evidences are very strong to support the conclusion.

As commented by the reviewer, we did not exclude the slight possibility that *Gdf1/3-like* was translocated close to *Lefty* with its own enhancers. Since the expressions of *Gdf1/3-like* and *Lefty* are highly similar but not identical (we have added this statement in the section “Gdf1/3-like is indispensable for body axes formation in amphioxus” of the revised manuscript). Besides those most cis-regulatory elements (such as enhancers), hijacked from *Lefty*, in the intergenic region, sequences from other regions like introns are probably responsible for *Gdf1/3-like* expression.

We performed double ISH for *Lefty* and *Gdf1/3-like* and confirmed that the two genes are co-expressed approximately at the dorsal side at blastula and gastrula stage and the left side at neurula stage as expected (Figure 1—figure supplement 5).

Can putative common enhancer or factor binding sites be identified by analysis of the intergenic sequence? Is there any Foxh1, Smad-responsive motifs? Are both Gdf1/3-like and Lefty expressions affected upon mutation of the intergenic sequence?

We did find several Smad2/3-responsive motifs in the intergenic sequence between *Lefty* and *Gdf1/3-like*. However, we think the current data is enough to support our conclusion and testing the activity of these sites is out of scope of the work. We would like to do this in our future study. We hope the reviewer would appreciate this.

Please comment on differences in Figure 4 between mcherry and endogenous gene expression, and differences between mcherry and egfp. Illustration of SB505124 treatment is lacking as well as control of the effect on Nodal signaling.

As we discussed above (Major point 4 of reviewer #3) and stated in the ms, the 4 kb region includes most, but probably not all cis-regulatory elements required for the expression of *Lefty* and *Gdf1/3-like*. Sequences from other regions like introns are probably responsible for endogenous *Lefty* and *Gdf1/3-like* expression. This could be the cause for the slight differences between *mCherry* and endogenous gene expression (Figure 4A)*.* We have revised the sentence “the transcription of *mCherry* was similar (although not identical) to that of endogenous *Gdf1/3-like* and *Lefty* with simultaneous initiation and D-V and L-R asymmetric expression pattern” in the revised manuscript.

Embryos in Figure 4 B were transient, but not stable transgenic embryos. Expression of reporter gene in transient transgenic amphioxus embryos is mosaic and varies very much among individuals (*Kikuta and Kawakami, 2009; Shi et al., 2018*).

Illustration of SB505124 treatment is lacking as well as control of the effect on Nodal signaling.

In Figure 4 B, the injected embryos were treated with DMSO or SB505124, from 4-cell stage to G1 stage. The left panels are representative embryos with different categories of expression (strong, mild, weak and absent) treated with DMSO or SB505124. The right histogram indicates the percentages of each categories in the group of DMSO or SB505124 treatment. The effect of SB505124 treatment on Nodal signaling was validated by the phenotype of embryos treated. We added this in a supplementary figure in the revised manuscript.

It is intriguing to see that in maternal Nodal mutant, Gdf1/3-like and Lefty expressions re-appears before zygotic Nodal. Does it imply that they have alternative inducers? This is also consistent with the partial effect of SB505124.

Yes, we also think there are other inducers responsible for the expressions of *Nodal*, *Gdf1/3-like* and *Lefty*. We would like to dissect this in future study.

Duboc, V. and T. Lepage (2008). "A conserved role for the nodal signaling pathway in the establishment of dorso-ventral and left-right axes in deuterostomes." *J Exp Zool B Mol Dev Evol* 310(1): 41-53.

Kikuta, H. and K. Kawakami (2009). "Transient and stable transgenesis using tol2 transposon vectors." *Zebrafish: Methods and protocols*: 69-84.

Kozmikova, I., S. Candiani, P. Fabian, D. Gurska and Z. Kozmik (2013). "Essential role of Bmp signaling and its positive feedback loop in the early cell fate evolution of chordates." *Dev Biol* 382(2): 538-554.

Li, G., X. Liu, C. Xing, H. Zhang, S. M. Shimeld and Y. Wang (2017). "Cerberus-Nodal-Lefty-Pitx signaling cascade controls left-right asymmetry in amphioxus." *Proc Natl Acad Sci U S A* 114(14): 3684-3689.

Onai, T., J. K. Yu, I. L. Blitz, K. W. Cho and L. Z. Holland (2010). "Opposing Nodal/Vg1 and BMP signals mediate axial patterning in embryos of the basal chordate amphioxus." *Dev Biol* 344(1): 377-389.

Opazo, J. C., S. Kuraku, K. Zavala, J. Toloza-Villalobos and F. G. Hoffmann (2019). "Evolution of nodal and nodal-related genes and the putative composition of the heterodimers that trigger the nodal pathway in vertebrates." *Evol Dev* 21(4): 205-217.

Opazo, J. C. and K. Zavala (2018). "Phylogenetic evidence for independent origins of GDF1 and GDF3 genes in anurans and mammals." *Sci Rep* 8(1): 13595.

Putnam, N. H., T. Butts, D. E. Ferrier, R. F. Furlong, U. Hellsten, T. Kawashima, M. Robinson-Rechavi, E. Shoguchi, A. Terry and J.-K. Yu (2008). "The amphioxus genome and the evolution of the chordate karyotype." *Nature* 453(7198): 1064-1071.

Satou, Y., S. Wada, Y. Sasakura and N. Satoh (2008). "Regulatory genes in the ancestral chordate genomes." *Dev Genes Evol* 218(11-12): 715-721.

Shen, M. M. (2007). "Nodal signaling: developmental roles and regulation." *Development* 134(6): 1023-1034.

Shi, C., J. Huang, S. Chen, G. Li and Y. Wang (2018). "Generation of two transgenic amphioxus lines using the Tol2 transposon system." *J Genet Genomics* 45(9): 513-516.

Soukup, V., L. W. Yong, T.-M. Lu, S.-W. Huang, Z. Kozmik and J.-K. Yu (2015). "The Nodal signaling pathway controls left-right asymmetric development in amphioxus." *EvoDevo* 6(1): 1-23.

Zhang, H., S. Chen, C. Shang, X. Wu, Y. Wang and G. Li (2019). "Interplay between Lefty and Nodal signaling is essential for the organizer and axial formation in amphioxus embryos." *Dev Biol* 456(1): 63-73.